# Diet, Obesity, and Depression: A Systematic Review

**DOI:** 10.3390/jpm11030176

**Published:** 2021-03-03

**Authors:** Olivia Patsalos, Johanna Keeler, Ulrike Schmidt, Brenda W. J. H. Penninx, Allan H. Young, Hubertus Himmerich

**Affiliations:** 1Department of Psychological Medicine, Institute of Psychiatry, Psychology & Neuroscience, King’s College London, London SE5 8AF, UK; olivia.patsalos@kcl.ac.uk (O.P.); johanna.keeler@kcl.ac.uk (J.K.); ulrike.schmidt@kcl.ac.uk (U.S.); allan.young@kcl.ac.uk (A.H.Y.); 2South London and Maudsley NHS Foundation Trust, London SE5 8AZ, UK; 3Department of Psychiatry, Amsterdam Public Health Research Institute, Amsterdam UMC, Vrije Universiteit, 1081 BT Amsterdam, The Netherlands; b.penninx@amsterdamumc.nl

**Keywords:** obesity, depression, diet, systematic review, weight loss

## Abstract

Background: Obesity and depression co-occur in a significant proportion of the population. Mechanisms linking the two disorders include the immune and the endocrine system, psychological and social mechanisms. The aim of this systematic review was to ascertain whether weight loss through dietary interventions has the additional effect of ameliorating depressive symptoms in obese patients. Methods: We systematically searched three databases (Pubmed, Medline, Embase) for longitudinal clinical trials testing a dietary intervention in people with obesity and depression or symptoms of depression. Results: Twenty-four longitudinal clinical studies met the eligibility criteria with a total of 3244 included patients. Seventeen studies examined the effects of calorie-restricted diets and eight studies examined dietary supplements (two studies examined both). Only three studies examined people with a diagnosis of both obesity and depression. The majority of studies showed that interventions using a calorie-restricted diet resulted in decreases in depression scores, with effect sizes between ≈0.2 and ≈0.6. The results were less clear for dietary supplements. Conclusions: People with obesity and depression appear to be a specific subgroup of depressed patients in which calorie-restricted diets might constitute a promising personalized treatment approach. The reduction of depressive symptoms may be related to immunoendocrine and psychosocial mechanisms.

## 1. Introduction

Both depression and obesity are major public health concerns [1,2] with high worldwide prevalence and associated increased cardiovascular risks [3,4]. Research has revealed an association between depression and obesity, with the prevalence of depression in obese individuals being twice as high as in those of normal weight [5]. The relationship between depression and obesity, although established and confirmed by numerous epidemiological studies and meta-analyses, has not yet been fully clarified. The association has been repeatedly examined with some authors asserting that depression results in weight gain and obesity and others claiming that obesity leads to depression, implying a bidirectional causality [6]. It has been suggested that both depression and obesity are due to dysregulation of stress responses, principally involving the hypothalamic–pituitary–adrenal (HPA) axis [7]. Additional mechanisms linking the two disorders are inflammation, oxidative stress, and other endocrine dysfunctions [8], as well as psychological mechanisms such as rumination, stigmatization and ostracism that contribute to and maintain the bidirectional relationship [9,10].

### 1.1. Diet and Depression

The typical diets of western societies have high amounts of saturated fats and refined sugars, as well as high amounts of red and processed meats, with concurrent low levels of fruit, vegetable and fiber intake. This results in a diet that is energy-dense and nutrient-poor with profound consequences for both our physical and mental health. The relationship between diet and obesity is clear; individuals consuming more calories than the recommended daily allowance, combined with consuming high amounts of foods high in fat and sugar content, are more likely to develop obesity. More recently, the impact of diet on mental health has also been revealed to be significant; for example, a recent meta-analysis found that adults following a healthy dietary pattern have fewer depressive symptoms and lower risk of developing depressive symptoms [11].

The precise etiology of depression is unknown, but many psychological, social, and biological underpinnings are thought to contribute to its development [12]. The latter includes genetic, hormonal, immunological, biochemical, and neurodegenerative factors. Concurrently, research has shown that these physiological aspects can be modulated by diet and nutrition. For example, in the case of genes, vitamin E has been shown to modulate several genes involved in neural signal transduction, inflammation and cell proliferation among others, while omega-3 polyunsaturated fatty acids (n-3 PUFAs) [13] have been shown to interact with genes that code for cytokines, cholesterol metabolizing enzymes, and growth factors [14].

### 1.2. Depression and Obesity

Many authors posit that depression is a heterogenous assortment of symptoms that can be divided into subtypes based on the accompanying presenting symptoms beyond low mood. Most recently, it has been subdivided into two main subtypes: type 1, which is characterized by loss of appetite and body weight, insomnia, and suicidal ideation, and type 2, also known as atypical depression, which presents with increased appetite and weight gain, leaden paralysis, hypersomnia, and a persistently poor metabolic profile [15]. Several factors are thought to moderate the relationship between obesity and depression. Stunkard et al. have reviewed the literature pertaining to what those moderators and mediators could be, and they have identified several including the severity of obesity, the severity of depression, and stress [16].

Correlations between both disorders involve disturbance of appetite regulation, changes in metabolic, hormonal and immunological parameters, and behavioral problems such as reduced physical activity [9,10,17,18,19,20]. More specifically, obesity has been shown to induce important physical, psychological, and behavioral changes in vulnerable patients, such as changes in the hormone and cytokine systems [18,21], changes in thought processes such as rumination [9], and behavioral changes such as reduced physical activity [20]. These changes are known risk factors of depression [17,20]. Thus, in obese patients, depression can be seen as a health consequence of obesity. If obesity contributes to the development and maintenance of depression, we can hypothesize that weight loss might help those depressed patients who are obese. Indeed, recent studies indicate that weight loss due to caloric restriction or gastric bypass surgery improves depressive symptoms among obese patients with depression [22,23,24,25]. Therefore, we sought to review and collate the existing research literature on the effects of diet modifications on depressive symptoms in overweight or obese individuals enrolled in dietary weight loss programs. The underlying idea was that in people with obesity and depression, depression occurs as a consequence of obesity, and therefore weight loss could not only help with regard to obesity but could also reduce depressive symptoms.

## 2. Materials and Methods

We conducted this systematic review according to the Preferred Reporting Items for Systematic Reviews and Meta-Analyses (PRISMA) guidelines [26].

### 2.1. Literature Search

Three electronic databases (PubMed, Medline, and EMBASE) were systematically searched from inception until 5 October 2020 using the following search terms: *diet* in combination with *depression*, in combination with *obesity*. Reference lists of potentially relevant papers and reviews were also scanned for potentially eligible papers.

### 2.2. Eligibility Criteria

Searches were limited to abstracts, studies with adult human participants, and studies written in English. Any study which assessed the effect of any dietary intervention or dietary supplementation on depressive symptoms in the context of obesity (BMI ≥ 30 kg/m^2^) at baseline and at least at one follow-up point was eligible for inclusion. To be eligible, at least a subgroup of study participants had to be obese. However, we did not exclude studies, when in addition to people with obesity, other study participants were overweight or of normal body weight (used as controls).

Studies were excluded if they (a) were not longitudinal clinical studies, (b) did not comment on weight/BMI change after intervention, (c) did not discuss change of depressive symptoms after intervention, or (d) were association or observational studies without a dietary intervention. Review articles, meta-analyses, case studies, conference proceedings/abstracts, book chapters, and unpublished theses were not included.

### 2.3. Study Selection

Figure 1 depicts the study selection and screening flowchart. Titles and abstracts of publications resulting from the search were imported into Mendeley and duplicates were removed. Two independent reviewers (O. P. and J. K.) performed all stages of the search, screening, and evaluation. Titles and abstracts were screened, and irrelevant articles were disregarded. Articles whose abstracts passed the first screen were read in full and assessed for eligibility based on our prespecified inclusion criteria, described above. Study quality assessment was performed using a quality assessment tool for pre-post studies from the National Heart, Lung and Blood Institute [27].

## 3. Results

### 3.1. Characteristics of Included Studies

Individual study characteristics are described in Table 1. Twenty-four longitudinal clinical trials met the inclusion criteria. A total of 3244 patients participated in trials investigating the impact of diet, dietary supplements, or behavioral modification/counselling on weight and depression scores. All study participants were overweight or obese aside from some of the participants in the Breymeyer et al. [28] study which included a group of nonobese participants used as healthy controls. 

The sample sizes of included studies ranged between n = 25 [29] and n = 1025 [30] participants, and adherence and completion rates varied between ≈60% [31] and 100% [28,32,33,34,35,36,37,38] (see Table 1). Mean age of patients was reported in 19 studies, with a combined mean of 47.1 years. Gender was reported in 21 studies with a total of 2041 females and 863 males. The mean BMI was reported in 15 studies and pooling those means gave a mean of 33.9 kg/m^2^. The shortest intervention duration was 28 days [28] whilst the longest was 52 weeks [31,39,40,41]. One study did not explicitly state the data collection end point [35].

The most frequently used depression scale was Beck’s Depression Inventory (BDI) used by 11 studies [25,31,32,33,35,39,40,42,43,44,45], followed by the Profile of Mood States (POMS) used in six [28,31,39,41,44,46], and the Centre for Epidemiologic Studies Depression Scale (CES-D) [28,36,38] and the Hospital Anxiety and Depression Scale (HADS) used in three [29,47,48]. Two studies used the 21-item Depression Anxiety Stress Scale (DASS-21) [34,49] A further seven scales were used by some studies, either in conjunction with the aforementioned, or on their own (see Table 1). The majority of studies compared different dieting therapy groups to each other, with only three studies comparing an energy-restricted dieting group to a nondieting control group [43,47,50].

**Table 1 jpm-11-00176-t001:** Characteristics of included studies.

Study	Disease	Sample Size (Recruited)	Excluded Due to Nonadherence to Intervention	Excluded or Withdrawn for Other Reasons	Completed	Diet Intervention	Energy Restricted Diet	Nondieting Control Group	Depression Scale	Gender (M)	Age (Mean ± SD)	Summary	Quality Assessment
Bot et al. [30]	Obesity	1025			779	Multinutrient supplementation + FRBA	No	No	MINI, PHQ-9	772 (253)	46.6	No significant effect of supplements or FRBA on PHQ scores.	Good
Breymeyer et al. [28]	Overweight/ Obese vs. healthy	82			82	Isocaloric HGL and LGL (crossover)	No	No	POMS, CES-D	41 (41)		Mood disturbance was higher on HGL diet. Significant effect of diet on CES-D score with higher depression score associated with HGL diet.	Good
Brinkwork, Buckley at al. [31]	Overweight/ Obesity	106	4		66	Energy restricted LCHF vs. HCLF	Yes	No	POMS, BDI		50 ± 0.8	Both diet groups achieved significant reduction in weight and depression scores. However, LC group rebounded to baseline levels over time whereas LF group depression scores remained low.	Good
Brinkworth, Luscombe-Marsh et al. [39]	Obesity + diabetes	115	6	32	77	Energy restricted LCHF vs. HCLF	Yes	No	POMS, BDI		58.5 ± 7.1	Both diet groups achieved significant decrease in weight, POMS, and BDI scores.	Good
Canheta et al. [47]	Obesity	149		36	113	Brazilian diet vs. extra virgin olive oil vs. both	Yes	Yes	HADS	109 (20)	38.9 ± 8.7	All diet groups achieved significant reduction in depression scores.	Good
Coates et al. [46]	Overweight/ Obese	151	2	20	128	Isocaloric AED vs. NF	No	No	POMS	78 (70)	65 ± 8	No reduction in weight or depression scores.	Good
Crerand et al. [43]	Obesity	123				Meal replacement or balanced deficit diet vs. control (nondieting group)	Yes	Yes	BDI	123 (0)		Diet group lost significantly more weight and reported significantly greater reduction in depressive symptoms.	Good
Fuller et al. [40]	Obesity	70			60	Diet + exercise (Korean vs. Western hypocaloric)	Yes	No	BDI-II	44 (36)	45.5 ± 11.1	Significant decrease for both groups in weight and BDI scores at end of intervention.	Good
Galletly et al. [29]	Overweight + PCOS	25				LPHC vs. HPLC	Yes	No	HADS	25 (0)	HPLC: 33 ± 1.2 LPHC: 32 ± 1.2	HPLC diet resulted in significant reduction in depression scores. No difference in weight loss between diet groups.	Good
Hadi et al. [49]	Overweight/ Obese	60	0	1	59	Synbiotics vs. placebo	No	N/A	DASS-21	20 (40)	Synbiotic: 34.5 ± 6 Placebo: 36.6 ± 7.3	Both groups showed decreased weight and depression scores, however, synbiotic group showed greater improvement compared to placebo.	Good
Halyburton et al. [44]	Overweight/ Obese	121	5	21	95	Energy restricted LCHF vs. HCLF	Yes	No	POMS, BDI	95 (0)	LCHF: 50.6 ± 1.1 HCLF: 49.8 ± 1.3	LCHF significantly greater weight loss than HCLF. Significant reduction in POMS and BDI scores for both diet groups.	Very good
Hariri et al. [33]	Overweight/ Obesity	62			62	Energy restricted diet plus sumac supplement vs. energy restricted diet + placebo	Yes	No	BDI-II	62 (0)	S: 42 ± 8.44 C: 44 ± 11.8	Significant reduction in weight and depression in both groups. Sumac supplement group showed significantly more reduction in weight.	Good
Lutze et al. [41]	Obesity	117	8	43	66	Isocaloric HP vs. HCLF	Yes	No	POMS, SF-36 mental health summary	0 (66)	49.6 ± 9.2	No effect of HP vs. HC diet. Both diets resulted in reduced weight and reduced POMS and SF-36 scores.	Good
Pedersen et al. [48]	Overweight/ Obesity	70			55	AIT vs. LED	Yes	No	HADS	12 (43)		LED mean weight loss: 9.9kg, AIT mean weight loss: 1.6%. No significant change in HADS.	Good
Raman et al. [34]	Obesity	80			80	BWL vs. BWL + CRT-O	No	N/A	DASS-21	69 (11)	CRT-O: 40.6 ± 7.0 C: 42.2 ± 8.8	BWL + CRT-O resulted in significantly more weight loss at 3-month follow-up but had no effect on depression scores	Good
Rodriguez-Lozada et al. [32]	Overweight/ Obese	305			305	MHP vs. LF	Yes	No	BDI	213 (92)	45.3	Both energy intake restricted diets resulted in reduced weight and depression scores. LF diet had more pronounced effects on depression scores in women.	Good
Ruusunen et al. [35]	Overweight/ Obese + impaired glucose tolerance	140			140	Counselling on weight reduction + physical activity	No	N/A	BDI	81 (59)	57.7 ± 6.4	Both groups achieved reductions in weight and depression scores. With participants showing the greatest reduction in weight also showing greater decreases in depression scores.	Good
Sanchez et al. [45]	Obesity	105			104	Moderate energy restriction + probiotic	Yes	No	BDI	60 (45)	35 ± 10	Significant decrease in BDI scores in probiotic group compared to placebo.	Good
Tan et al. [50]	Overweight/ Obesity + insomnia	73	2	6	49	Energy restricted diet vs. control	Yes	Yes	Rimon’s brief depression scale	0 (49)	D: 51 C: 52.6	Diet group improved sleep time and depression scores. However, depression scores reduced in both groups.	Good
Uemura et al. [36]	Obesity	44			44	Counselling on gut microbiota	No	N/A	CES-D	44 (0)	I: 62 ± 8.7 C: 63.3 ± 9.1	BMI, body weight, and CES-D scores decreased significantly after intervention.	Good
Vaghef-Mehrabany et al. [25]	Obesity + MDD	62	6	11	45	25% weight loss diet + probiotic vs. placebo	Yes	No	BDI-II, HDRS	62 (0)		Regardless of supplementation group, patients who achieved >1.9kg reduction in weight, showed reduction in HDRS and borderline reduction in BDI-II. Prebiotic supplementation had no effect on depressive symptoms.	Good
Vigna et al. [37]	Overweight/ Obese	77			77	LCD: *Hericium erinaceus* vs. control	Yes	No	Zung’s depression scale, SCL-90	65 (12)	53.2 ± 0.7	*H. erinaceus* supplementation decreased depression scores.	Very good
Webber et al. [38]	Overweight/ Obese	49			49	BWL vs. EBT	No	N/A	CES-D	41 (8)	45 ± 7.9	Both groups showed improvements in BMI and depression scores.	Good
Wing et al. [42]	Obesity + diabetes	33		2	31	VLCD vs. balanced diet	Yes	No	BDI	25 (18)		Both weight and BDI scores decreased significantly after intervention. VLCD group had more weight loss.	Fair

Abbreviations: AED = almond-enriched diet, AIT = aerobic interval training, BDI = Beck’s depression inventory, BDI-II = Beck’s Depression Inventory-2, BWL = behavioral weight loss, CES-D = center for epidemiologic studies depression scale, CRT-O = cognitive remediation therapy for obesity, DASS-21 = depression anxiety stress scale 21 items, EBT = emotional brain training, FRBA = food-related behavioral activation, HADS = hospital anxiety and depression scale, HCLF = high carbohydrate and low fat diet, HDRS = Hamilton depression rating scale, HGL = high glycemic index, HP = high protein diet, HPLC = high protein, low carbohydrate diet, LCD = low calorie diet, LCHF = low carbohydrate, high fat diet, LED = low energy diet, LF = low fat diet, LGL = low glycemic index, LPHC = low protein, high carbohydrate diet, MHP = moderately high protein diet, MINI = mini international neuropsychiatric interview, N/A = not applicable, NF = nut-free diet, PHQ-9 = patient health questionnaire, POMS = profile of mood states, SD = standard deviation, VLCD = very low calorie diet.

### 3.2. Study Findings

A summary of the findings of each of the included studies can be found in Table 2. Not all studies provided all values for every outcome measure but all of them commented on the desired outcomes, i.e., the effects of diet interventions on depressive symptoms in obese or overweight participants. Overall, the majority of studies concluded that weight loss, whether through calorie restriction, dietary supplements, or behavioral training, resulted in a reduction of depressive symptoms, with reported values of effect sizes on depression and depressive symptoms varying between a Cohen’s d of 0.16 [42] and 0.64 [49], while effect sizes of weight change ranged from a Cohen’s d of 0.0 [31] to 0.45 [39].

As we obtained studies in people with obesity and diagnosed depression and with obesity and depressive symptoms without the clinical diagnosis of depression, we will report on these two types of studies in separate sections. In people with obesity and depressive symptoms, but no diagnosis of depression, authors used different treatment approaches: energy restricted diets, energy restricted diets plus pre/probiotic supplementation, diet combined with an exercise intervention, and counselling. Thus, we dedicated one paragraph to each of these approaches. However, as these are not disjointed categories, some studies fell into multiple categories.

#### 3.2.1. Effects of Diet Interventions on Obesity and Clinically Diagnosed Depression

Of the included studies only three were conducted in participants with concurrent obesity and clinically established depressive disorder. Participants in these three studies were on an energy restricted diet plus an additional supplement or placebo. Hariri et al. reported all relevant values for weight, BMI, and depression scores and demonstrated a decrease in weight and depression scores for both groups (sumac vs. placebo) [33]. Vaghef-Mehrabany et al. [25] and Vigna et al. [37] did not provide values for all groups at all time points but nonetheless commented on the outcomes. Vaghef-Mahrabany et al. did not find any difference between the group receiving supplementation and the placebo group, however, they did note that regardless of group classification, participants that lost more than 1.9 kg of weight showed significantly improved depression scores. In contrast, Vigna et al. reported significant reductions in depression scores for the group receiving the *H. erinaceus* supplement.

#### 3.2.2. Effects of Diet Interventions on Obesity and Depressive Symptoms

##### Studies of Energy Restricted Diets

The majority of studies (*n* = 16) investigated the effects of specific calorie restricted diets on weight and depressive symptoms in overweight or obese participants without an established current clinical diagnosis of depression. None of the studies included here reported full datasets with values at each time point and corresponding significance values. Most authors reported a decrease in depressive symptoms following a calorie restricted diet, aside from one study [48] that reported no change in depression scores. Three studies compared a calorie restricted diet with a noncalorie restricted control group, and all three found a reduction in both weight and depression scores in the intervention group [40,43,50]. However, most studies compared different calorie reduced diets with each other, i.e., four studies compared a low carbohydrate, high fat (LCHF) diet with a high fat, low carbohydrate (HCLF) diet [29,31,39,44], one study compared a high protein diet with a high carbohydrate diet [41], one study compared a high protein diet with low fat diet [32], one compared a very low calorie diet with an energy reduced balanced diet [42], and one study compared the traditional Brazilian diet with olive oil supplementation [47]. The remaining studies on energy restricted diets included the use of dietary supplements and will thus be discussed separately in the next section.

##### Studies on Energy Restricted Diets Plus Pre/Probiotic Supplementation

Three studies reported on the impact of calorie restriction with additional pre/probiotic supplementation [25,45,49]. Hadi et al. [49] and Sanchez et al. [45] found a significant decrease in depression scores for the groups receiving pro/prebiotics whereas Vaghef-Mehrabany et al. [25] reported a decrease in depression scores for participants who achieved a weight loss greater than 1.9kg regardless of prebiotic supplementation.

##### Studies on Diet Combined with Exercise Intervention

Four studies investigated the impact of diet and exercise/lifestyle interventions on depressive symptoms [35,39,40,48]. Three of them reported significant reductions in depression scores accompanying reductions in weight, except from Pedersen et al. [48] who reported no differences in depression scores between the two groups (aerobic interval training (AIT) vs. low energy diet (LED)) even though the LED group achieved a 10.4% decrease in body weight.

##### Studies on Counselling (Not Explicitly Calorie Restricted)

We found seven studies that reported on the effects of supplements without calorie restriction [46] or on the effects of behavioral modifications/counselling on depression scores [28,30,34,35,36,38]. These studies were not specifically prescribing calorie restricted diets but were rather providing additional supplements and/or counselling on healthy lifestyle modifications, such as dietary and exercise recommendations. The exception to this was the Breymeyer et al. study, which did not include any training or calorie restriction but was comparing the effects of a high glycemic (HG) diet (vs. a low glycemic (LG) diet) on depression scores [28]. The authors concluded that mood disturbance was higher for the group on the HG diet, with higher depression scores associated with higher glycemic load. Coates et al. investigated the effects of an almond-enriched diet compared to a nut-free diet and found no differences in depression scores between the two groups [46]. Three studies [35,36,38] investigated what effect counselling or behavioral training has on depression scores and all three found improvements in depressive symptoms following the intervention. One study investigated the effects of multinutrient supplementation and/or food-related behavioral activation therapy (in a 2 × 2 design) on depressive symptoms and did not find any significant effect of either on depression scores [30]. Lastly, one study compared the effect of behavioral weight loss with or without cognitive remediation therapy on body weight and depression scores and found no changes in depression scores between the two groups [34].

## 4. Discussion

### 4.1. Summary of the Main Findings

This systematic review summarizes the existing data on the effects of diet on depressive symptoms in overweight or obese patients. Findings from the included studies were mixed, with the majority of studies reporting significant improvements in depression scores after diet and weight loss, and the remaining studies reporting no differences between depression scores between pre- and postintervention [34,46,48]. No studies reported deterioration of depressive symptoms aside from one that reported increased mood disturbance in participants on a high glycemic load diet [28]. Importantly, the majority of authors reported high adherence to the intervention, whether those were hypocaloric diets or supplements. The trend of obese individuals experiencing an improvement in their depressive symptoms after diet and weight loss is in line with previous research. Dietary interventions using a calorie-restricted diet (e.g., [25,29,44]) resulted in decreases in depressive symptoms. However, the results are less clear for dietary supplements (e.g., [33,37,46]). Overall, the dietary approaches were heterogenous in that the diets investigated were calorie reduced, traditional, high/low in protein, high/low in carbohydrates, with/or without pre-/probiotic, vitamins, or naturopathic supplements.

### 4.2. Possible Mechanisms for Improved Mood after Weight Loss

It is well established that depression and obesity co-occur to a high degree [5,6,51,52,53,54], however the relationship between the two disorders is complex and currently of ambiguous directionality. Stunkard et al. presented a summary on the existing data using a moderator/mediator framework in which they classified eating and physical activity as an important mediator of obesity and comorbid depression [16]. Some authors consider depression as a consequence of obesity resulting from societal stigmatization, dissatisfaction with one’s appearance, and low self-esteem [55,56,57]. Others consider obesity as resulting from decreased physical activity, excessive ‘comfort’ eating, and antidepressant medication use that often accompanies depression [58,59,60,61,62].

Several epidemiological studies have found associations between mood and diet. Particularly, a western-style diet high in processed foods and sugar content and low in fruits and vegetables, is associated with worsening of mood states. Indeed, one of our included studies found increases in depression scores in participants on a high glycemic load diet [28]. Diets that are high in carbohydrates but low in fat and protein have also been associated with lower mood scores in cross-sectional studies [63,64], whereas an abundance of research extols the beneficial effects of Mediterranean-style diets [65] which are high in fruit, vegetables, nuts, pulses and wholegrains, low in fat and carbohydrate, with very little processed foods. The differences in mood scores between these two types of diets are thought to be partly due to the increased systemic inflammation and oxidative processes that often accompanies a western-style diet [66,67,68,69].

#### 4.2.1. Physiological Mechanisms

Research has exposed metabolic and inflammatory dysregulation as a common denominator in depression and obesity [70,71]. Additionally, both depressed and obese patients exhibit dysregulation of the hypothalamic–pituitary–adrenal (HPA) axis [72,73] and consequently chronic elevations in cortisol [74,75]. Increases in cortisol levels have been reported as having a causal role in depression, as well as leading to weight gain, specifically in abdominal adiposity. Recently, white adipose tissue (WAT) has been conceptualized as an endocrine organ, as opposed to how it was previously thought of—as an inert storage tissue—due to its ability to produce cytokines and other related molecules. Among these are interleukin (IL)-1β, IL-6, and tumor necrosis factor (TNF)-α [76,77,78], which are known proinflammatory cytokines, as well as chemokines, including monocyte chemoattractant protein (MCP)-1 [79,80]. The ensuing signaling cascade leads to immune activation and white blood cell accumulation, and an overall increased inflammatory response. This immune activation has various downstream effects. For example, IL-2 reduces tryptophan plasma levels [81], possibly by activating tryptophan 2,3-dioxygenase (TDO) and indoleamine 2,3-dioxygenase (IDO). Tryptophan is an essential amino acid necessary for 5-HT synthesis. Low levels of tryptophan could lead to lower levels of serotonin and thus affect mood. Another example is the accumulation of peripheral monocytes in the brain as a result of systemic inflammation [82], and specifically the increased production of MCP-1 in hypothalamic neurons. This monocyte migration has been associated with increased anxiety and depression [83]. Minimization of accumulated adipose tissue through weight loss could attenuate this inflammatory process, leading to improved mood.

Another molecule implicated in both obesity and depression is leptin. Leptin is a peptide hormone released by adipocytes and crosses the blood–brain barrier via a saturable transport mechanism. Low plasma levels of leptin have been observed in depressed patients [84,85]. In the case of obesity however, plasma leptin levels have been found to be elevated [86,87]. This contradictory finding can be explained by leptin resistance (as in the case of type 2 diabetic patients being resistant to insulin) and could be a result of impaired transport across the blood–brain barrier, of reduced function of the leptin receptor, or errors in signal transduction [88,89]. Similar to cortisol and inflammatory molecules described previously, leptin modulates HPA axis function [90,91]. Leptin also interacts with monoamines and although its effect on monoamine neurotransmission remains unclear, there is evidence for leptin’s involvement in the 5-HT system [92] and in the activation of STAT3 in dopamine neurons of the ventral tegmental area (VTA) [93]. Reducing the amount of adipose tissue through diet and subsequent weight loss could ameliorate leptin resistance, reinstate leptin function, and relieve low mood.

#### 4.2.2. Psychosocial Mechanisms

It should be borne in mind that psychosocial attributes may affect physiology, and the distinction between the two mechanisms here is for ease of discussion. A good example of this environment x biology interaction is the finding that weight discrimination, often experienced by obese individuals, increases cortisol levels [94]. Additionally, repeated discrimination can lead to lower self-esteem and increased negative affect [95]. Many studies have reported on the negative attitudes of employers, peers, and even clinicians towards obese persons [96,97]. Continued maltreatment can impact obese persons’ mood and self-concept, both of which can contribute to depression.

Even if obese individuals do not experience weight discrimination or stigma by others, their self-esteem could be impacted by their own body image dissatisfaction (BID). Some research has found correlations between BID and depressive symptoms and suggested that obesity confers risk for developing depression through increased BID [98,99]. Therefore, it is possible that losing weight improves body image satisfaction and low mood. For a more thorough discussion see Markowitz et al. [100].

It is important to note that some researchers posit that while obese individuals experiencing weight loss also experience an improvement in mood, this improvement does not seem to be mediated by the weight loss itself but is rather related to active participation in treatment [101,102,103,104].

### 4.3. Clinical Implications

Given the high prevalence of obesity and depression and the strain exerted on healthcare systems it would be of great value if prescribing dietary modifications for the amelioration of obesity had the additional consequence of improving depressive symptoms. Our findings suggest that dietary interventions leading to weight loss improve mood scores in both clinically and subclinically depressed obese individuals. Importantly, adherence to intervention seemed to be high in our included studies, which provides clinicians reason for optimism.

People with obesity and depression or depressive symptoms are a particularly vulnerable group who are at risk of worsening of depressive symptoms (e.g., [105]), switching from depression to mania (e.g., [106]), and of the appearance of eating disorder symptoms (e.g., [107]). Thus, further studies in obese and depressed patients should focus on the safety of diets regarding the reoccurrence of depressive symptoms, the switch from depression to mania, and the appearance of eating disorder symptoms.

### 4.4. Strengths and Limitations

This is the first review to systematically collate research on the effects of dietary interventions on depression and depressive symptoms in overweight/obese patients. Our strict inclusion of longitudinal clinical trials strengthens the validity of our findings. Additionally, the quality of most of the studies was good, and only one was deemed fair (see Table 1). However, the respective study quality was deemed good according to each study’s specific research question which is not the same as being of good quality to answer the research question of this review. Therefore, our finding of weight loss ameliorating depression scores in obese individuals is based on limited and heterogeneous data.

Furthermore, even though a meta-analytic approach would have provided more quantifiable evidence, such an approach would have been inappropriate based on the heterogeneity of the studies. This heterogeneity emerged from both the plethora of dietary approaches investigated as well as the varied comparison groups, and the lack of data. However, future meta-analytic research could investigate well-defined dietary categories by being less stringent with inclusion criteria, for example by including all studies in depressed patients regardless of the weight status.

A further limitation of our review is the inclusion of only three studies that compared the depression scores of participants in an energy-restricted diet group to a non-dieting control group. The lack of well-defined randomized controlled trials (RCTs) with this specific research question limits the validity and generalizability of our conclusion. Further RCTs are necessary to confirm the trend we have noted in this review.

Our study focused on the use of diet in people with both, obesity and depression. We did not include studies if obesity was not an important aspect of the study design, e.g., the SMILES trial [108] and the HELFIMED study [109], both of which showed that dietary improvement is associated with a reduction in depression scores. However, this systematic review focused on people with both, obesity and depression, because we wanted to investigate whether dietary modifications would help people who suffer from both disorders.

## 5. Conclusions

The findings of the current review provide preliminary evidence for the importance of weight loss in obese individuals experiencing low mood. The majority of studies included showed decreases in depression scores following dietary interventions, specifically through calorie-restricted diets. This is in line with a large body of research reporting amelioration of depressive symptoms in obese patients after weight loss. It is plausible that pursuing dietary interventions for obese patients with comorbid depression could have the additional benefit of relieving some of their depressive symptoms as well as improving their metabolic profile and cardiovascular risk. Therefore, a restricted diet might specifically help people with type 2 depression which is characterized by an increased appetite and weight gain, leaden paralysis, hypersomnia, and a persistently poor metabolic profile [13].

In summary, people with obesity and depression appear to be a specific subgroup of depressed patients. In this subgroup, calorie-restricted diets could constitute a promising personalized treatment approach which might lead to a reduction of depressive symptoms. The underlying mechanisms at play may be related to the immune and endocrine systems and to psychosocial aspects obesity.

## Figures and Tables

**Figure 1 jpm-11-00176-f001:**
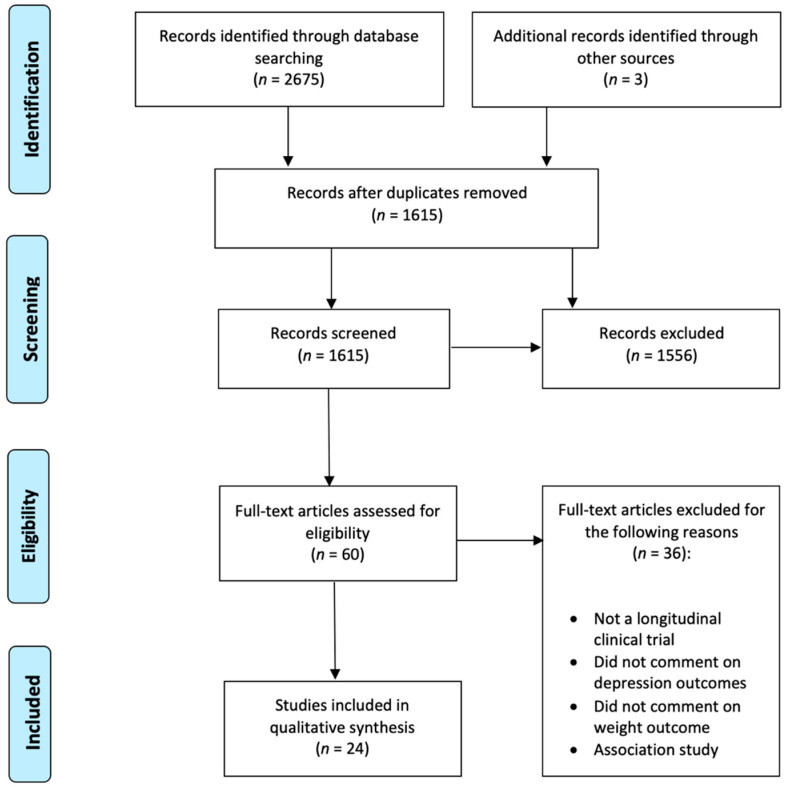
PRISMA flow diagram.

**Table 2 jpm-11-00176-t002:** Findings of included studies.

Study	Weight kg (Mean ± SD)			BMI kg/m^2^ (Mean ± SD)			Depression		
	Baseline	Post	*p*-Value	Baseline	Post	*p*-Value	Baseline	Post	*p*-Value
Bot et al. [30]				P: 31.4 P + FRBA: 31.2 S: 31.3 S + FRBA: 31.7			P: 7.3 (4.1) P + FRBA: 7.4 (4.4) S: 7.9 (4.4) S + FRBA: 7.1 (4)		
Breymeyer et al. [28]								HGL: 2.80 LGL: 2.03	***p* = 0.002**
Brinkworth, Buckley et al. [31]	LCHF: 96 ± 1.6 HCLF: 97.6 ± 1.6	LCHF: 82.3 ± 2.1 HCLF: 83.9 ± 1.9							BDI: ***p*** **= 0.05** POMS: ***p* = 0.05**
Brinkworth, Luscombe-Marsh et al. [39]	LC: 101.8 ± 2 HCLF: 101.1 ± 2	LC: 92.6 ± 2 HCLF: 91 ± 2							
Canheta et al. [47]				46.3 ± 6.5		***p* < 0.001**			
Coates et al. [46]	AED: 84.4 ± 12 NF: 85.4 ± 14	AED: 84.8 ± 1.38 NF: 85.6 ± 1.36	*p* > 0.05	AED: 30.2 ± 0.44 NF: 30.6 ± 0.43	AED: 30.5 ± 0.44 NF: 30.3 ± 0.43	*p* > 0.05	AED: 0.89 ± 1.9 NF: –3.74 ± 1.88	AED: 1.11 ± 2.2 NF: –2.22 ± 2.17	POMS: *p* > 0.05
Crerand et al. [43]	D: 97.8 ± 13.5 C: 96.1 ± 12.1			D: 36.2 ± 4.5 C: 35.3 ± 4.3		D vs. C: ***p*** **< 0.001**	D: 7.7 ± 5.5 C: 7.4 ± 5.9		***p* < 0.001**
Fuller et al. [40]	D: 90.9 ± 12.2 C: 93.8 ± 12.7	D: –7.9 ± 2.1 C: 0.1		D: 34.1 ± 4.3 C: 35.2 ± 4.8			D: 22.1 ± 8.1 C: 23.7 ± 11.1	D: 19.3 ± 6 C: 25.3 ± 12.7	POMS: time x group ***p* < 0.001**
Galletly et al. [29]	HPLC: 104.2 ± 5.3 LPHC: 98.6 ± 4.6	HPLC: –6.9 ± 0.8 LPHC: –8.5 ± 6.3		HPLC: 37.6 ± 6.4 LPHC: 37.2 ± 6.9	HPLC: 34.5 ± 5.7 LPHC: 34.5 ± 6.3		HPLC: 5.6 ± 3.2 LPHC: 4.8 ± 3.4	HPLC: 3.6 ± 2.8 LPHC: 3.4 ± 3.3	HPLC: ***p* < 0.001** LPHC: NS
Hadi et al. [49]	89.4 ± 16.1	–5.2% ± 4.3%	***p* < 0.001**	31.1 ± 3.9			5 ± 4.6	2 ± 4.1	***p* < 0.001**
Halyburton et al. [44]	LCHF: 93.6 ± 2.1 HCLF: 97 ± 2.1			LCHF: 33.3 ± 0.6 HCLF: 33.8 ± 0.6					***p* < 0.001**
Hariri et al. [33]	Su: 84.3 ± 11.7 P: 79.3 ± 11.4	Su: 78.96 ± 10.84 P: 76.89 ± 11.35	***p* < 0.001**	Su: 32.4 ± 3.73 P: 31.2 ± 3.87	S: 30.4 ± 3.55 P: 30.3 ± 3.89	***p* < 0.001**	Su: 25.4 ± 9.42 P: 26.17 ± 11.21	Su: 25.4 ± 9.42 P: 26.17 ± 11.21	***p* < 0.001**
Lutze et al. [41]	HP: 100.5 ± 1.8 HC: 102.6 ± 1.8	HP: –12.3 ± 1.4 HC: –10.9 ± 1.4					HP: 23.4 ± 1.09 HC: 23.04 ± 1.05	HP: 20.77 ± 0.97 HC: 20.19 ± 0.94	POMS: ***p* < 0.001** SF-36 subscales vitality and mental health: ***p* < 0.001**
Pedersen et al. [48]	Median: 92.8		LED: ***p* < 0.001**	Median: 31.4					
Raman et al. [34]				CRT-O: 40.3 ± 7.8 C: 39.2 ± 7.4	CRT-O: 38.9 ± 7.6 C: 39.7 ± 8.4		CRT-O:19.1 ± 11.2 C: 13.3 ± 12.2	CRT-O: 4.5 ± 5.1 C: 15.4 ± 12.2	
Rodriguez-Lozada et al. [32]	87.7	–8.6	***p* < 0.001**	31.6	–3.1	***p* < 0.001**	6.6	–2.7	***p* < 0.001**
Ruusunen et al. [35]		–3.14 ± 4.5		30.5 ± 3.4	–1.16 ± 1.74	I vs. C: ***p* = 0.024**	I: 6.8 ± 5.6	I: –0.9 ± 4.5	I: ***p*** **= 0.03**
Sanchez et al. [45]	Pro: 95.1 ± 13.9	Pro: –5.3 ± 4.3		Pro: 33.8 ± 3.3			Pro: 4.4 ± 4.1	Pro: –1.5 ± 3	***p* < 0.05**
Uemura et al. [36]	I: 66.3 ± 8.74	I: 64.6 ± 8.07	***p* < 0.001**	I: 27.8 ± 3.1	I: 27.1 ± 2.82	***p* < 0.001**	I: 17.64 ± 13.58	I: 10.05 ± 7.4	***p* < 0.001**
Tan et al. [50]	D: 93.8 C: 93.1	D: 92.7 C: 94.4	D: ***p* < 0.05**	D: 29.4 C: 29.2			D: 5.0 C: 4.0	D: 4.0 C: 3.0	***p* < 0.05**
Vaghef-Mehrabany et al. [25]							For >1.9kg weight loss: HDRS: 13.2 BDI: 19.5	For >1.9kg weight loss: HDRS: 9.1 BDI: 14.7	For >1.9kg weight loss: HDRS: ***p* < 0.001** BDI: ***p* = 0.006**
Vigna et al. [37]				HE: 33.1 ± 0.84 C: 33.4 ± 0.83	HE: 32.01 ± 0.82 C: 32.08 ± 0.88		HE: 48.8 ± 1.03	HE: 43.2 ± 2.38	HE: ***p*** **< 0.05**
Webber et al. [38]	BWL: 99 ± 16.7 EBT: 101 ± 10.8			BWL: 36 ± 4.3 EBT: 37 ± 4.9	BWL: –1.3 EBT: –0.6	BWL: ***p* < 0.001** EBT: ***p* = 0.032** BWL vs. EBT: ***p*** **< 0.03**	BWL: 7.5 ± 6.4 EBT: 10.4 ± 9.8	BWL: –2.9 EBT: –3.1	BWL: ***p* = 0.012** EBT: ***p* = 0.006**
Wing et al. [42]	103.2 ± 16.9						VLCD: 14.6 ± 9.4 BD: 11.4 ± 7.2	VLCD: 5 ± 6.3 BD: 2.9 ± 2.8	VLCD: ***p*** **< 0.001** BD: ***p*** **< 0.001**

Abbreviations: AED = almond-enriched diet, BD = balanced diet, BDI = Beck’s depression inventory, BDI-II = Beck’s Depression Inventory-2, BWL = behavioral weight loss, C = control, CES-D = center for epidemiologic studies depression scale, CRT-O = cognitive remediation therapy for obesity, D = diet, EBT = emotional brain training, FRBA = food-related behavioral activation, HCLF = high carbohydrate and low fat diet, HDRS = Hamilton depression rating scale, HE = *H. erinaceus* supplement, HGL = high glycemic index, HP = high protein diet, HPLC = high protein, low carbohydrate diet, I = intervention group, LCHF = low carbohydrate, high fat diet, LED = low energy diet, LGL = low glycemic index, LPHC = low protein, high carbohydrate diet, NF = nut-free diet, P = placebo, Pro = probiotic group, POMS = profile of mood states, S = supplements, Su = sumac supplement group, SD = standard deviation, SF-36 = short form health status survey, VLCD = very low calorie diet.

## Data Availability

Not applicable.

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
