# Peer review of "Diet, Obesity, and Depression: A Systematic Review"

_jpm, 2021, doi:10.3390/jpm11030176_

Round 1

Reviewer 1 Report

This is well written systematic review aimed to ascertain whether weight loss through dietary interventions has the additional effect of ameliorating depressive symptoms in obese patients. The authors followed the recommender methodology for systematic reviews and found that the majority of studies show decrease in depression scores following dietary interventions. This is a clinically relevant message and finding that should encourage health care managers to improve the current standard of care in depression

Author Response

We thank reviewer 1 for their favourable review.

Reviewer 2 Report

Patsalos et al. wrote a clear systematic review which summarizes the available evidence on diet, obesity and depression well.

My main problem with this manuscript is in the discussion and conclusion. In my opinion, the discussion should be more critical on the quality of the included studies and the conclusion is too strong. Although many studies are of good quality, they are of good quality for the research question that they aim to answer. That research question is in many cases different from the research question of this review and most studies are pre-post studies without a control group. The conclusion is focused on weight loss and improvement of depressive symptoms and although I agree that the results of the studies go in that direction I also think it should be stated more careful. The amount of evidence on which this is based is very limited. Only three studies, with in total 266 patients, looked purely at a weight loss diet. Of these study it seems that two compared the change to a non-calorie restricted control group, but one of these did not show a significant difference between the intervention group and the control group. The authors mention briefly in the discussion that taking part in the study may have caused the effects, but I do find that insufficient. Thus, although the evidence that is summarized well here goes in the direction of a beneficial effect of weight loss on depressive symptoms I think that it is overstated to say that this is based on good quality evidence. Maybe it should be made clearer that there are very few real RCTs available on this subject and why that is the case, but also make clear that that is a weakness of the study.

Minor points:

Line 46: increasingly, please delete. In many Western countries it has plateaued or is even decreased.

Line 147-149: I assume this needs to be deleted?

In section 3.1 I miss a remark that many studies did not compare with a control group (no diet), but compared different dietary therapies, which makes it difficult to simply determine the effect of energy restriction.

Table 1 does not make clear which studies used a control group and which only compared different diet strategies. It should also be made clear whether the interventions were energy restrictive or not, for example in the study of Canheta et al. : are the Brazilian diet or the virgin olive oil diet energy restrictive?

Table 1. Crerand et al. I do not understand the sentence: “Both dieting groups experienced reduction in depressive symptoms but dieting group reported significantly greater reduction in depressive symptoms.”

 In line 282 and 283 you refer to reference number 63 and 64, which are actually two cross-sectional analyses with only 38 subjects each….. Rather weak studies.

Author Response

  • My main problem with this manuscript is in the discussion and conclusion. In my opinion, the discussion should be more critical on the quality of the included studies and the conclusion is too strong. Although many studies are of good quality, they are of good quality for the research question that they aim to answer. That research question is in many cases different from the research question of this review and most studies are pre-post studies without a control group. The conclusion is focused on weight loss and improvement of depressive symptoms and although I agree that the results of the studies go in that direction, I also think it should be stated more careful. The amount of evidence on which this is based is very limited. Only three studies, with in total 266 patients, looked purely at a weight loss diet. Of these studies it seems that two compared the change to a non-calorie restricted control group, but one of these did not show a significant difference between the intervention group and the control group.

We thank reviewer 2 for this comment, and we agree with these concerns. We have rewritten parts of our discussion and conclusion sections. All changes are tracked in the manuscript and the most significant is presented here:

In section 4.4. of the revised manuscript we are now writing: “However, the respective study quality was deemed good according to each study’s specific research question which is not the same as being of good quality to answer the research question of this review. Therefore, our finding of weight loss ameliorating depression scores in obese individuals is based on limited and heterogeneous data.

Furthermore, even though a meta-analytic approach would have provided more quantifiable evidence, such an approach would have been inappropriate based on the heterogeneity of the studies. This heterogeneity emerged from both the plethora of dietary approaches investigated as well as the varied comparison groups, and the lack of data. However, future meta-analytic research could investigate well-defined dietary categories by being less stringent with inclusion criteria, for example by including all studies in depressed patients regardless of the weight status.

A further limitation of our review is the inclusion of only three studies that compared the depression scores of participants in an energy-restricted diet group to a non-dieting control group. The lack of well-defined randomized controlled trials (RCTs) with this specific research question limits the validity and generalizability of our conclusion. Further RCTs are necessary to confirm the trend we have noted in this review.”

  • Line 46: Please delete “increasingly.” In many Western countries it has plateaued or is even decreased.

We have deleted the word “increasingly”. The respective sentence of the revised version reads: “The typical dietary patterns in western societies have high amounts of saturated fats and refined sugars, as well as high amounts of red and processed meats, with concurrent low levels of fruit, vegetable and fibre intake.”

  • Line 147-149: I assume this needs to be deleted?

That sentence was indeed left in error. We have now deleted it.

  • In section 3.1 I miss a remark that many studies did not compare with a control group (no diet), but compared different dietary therapies, which makes it difficult to simply determine the effect of energy restriction.

We have included a remark about the lack of a non-dieting control group in the majority of studies in section 3.1. and in section 4.4. In section 3.1. “Characteristics of included studies” we are now writing in our revised manuscript: “The majority of studies compared different dieting therapy groups to each other, with only three studies comparing an energy-restricted dieting group to a non-dieting control group [47,43,50].” In section 4.4. “Strengths and limitations”, we added: “A further limitation of our review is the inclusion of only three studies that compared the depression scores of participants in an energy-restricted diet group to a non-dieting control group.”

  • Table 1 does not make clear which studies used a control group and which only compared different diet strategies. It should also be made clear whether the interventions were energy restrictive or not, for example in the study of Canheta et al.: are the Brazilian diet or the virgin olive oil diet energy restrictive?

We have now amended Table 1 and included a column on energy restriction and another on whether there was a non-dieting control group.

  • Table 1. Crerand et al. I do not understand the sentence: “Both dieting groups experienced reduction in depressive symptoms but dieting group reported significantly greater reduction in depressive symptoms.”

We have now amended the sentence for clarity by writing “Diet group lost significantly more weight and reported significantly greater reduction in depressive symptoms.”

  • In line 282 and 283 you refer to reference number 63 and 64, which are actually two cross-sectional analyses with only 38 subjects each. Rather weak studies.

We have added the information that these are cross-sectional studies in the revised manuscript.

We thank reviewer 2 for their excellent comments. The changes have led to more clarity and more specific information about the included studies. We also agree with the reviewer that a more cautious interpretation of the findings is appropriate. Overall, we feel that the raised comments and concerns have helped to improve the manuscript significantly.

Round 2

Reviewer 2 Report

I am satisfied with the responses of the authors and agree with the changes that are made in the paper.

This manuscript is a resubmission of an earlier submission. The following is a list of the peer review reports and author responses from that submission.